# Quantitative Analysis and Monitoring of EZH2 Mutations Using Liquid Biopsy in Follicular Lymphoma

**DOI:** 10.3390/genes11070785

**Published:** 2020-07-13

**Authors:** Ákos Nagy, Bence Bátai, Alexandra Balogh, Sarolta Illés, Gábor Mikala, Noémi Nagy, Laura Kiss, Lili Kotmayer, András Matolcsy, Donát Alpár, Tamás Masszi, András Masszi, Csaba Bödör

**Affiliations:** 1MTA-SE Lendület Molecular Oncohematology Research Group, 1st Department of Pathology and Experimental Cancer Research, Semmelweis University, 1085 Budapest, Hungary; batai.bence@med.semmelweis-univ.hu (B.B.); nagy.noemi@med.semmelweis-univ.hu (N.N.); kiss.laura1@med.semmelweis-univ.hu (L.K.); kotmayer.lili@med.semmelweis-univ.hu (L.K.); matolcsy.andras@med.semmelweis-univ.hu (A.M.); alpar.donat@med.semmelweis-univ.hu (D.A.); 23rd Department of Internal Medicine, Semmelweis University, 1088 Budapest, Hungary; balogh.alexandra@med.semmelweis-univ.hu (A.B.); illes.sarolta@med.semmelweis-univ.hu (S.I.); masszi.tamas@med.semmelweis-univ.hu (T.M.); masszi.andras@med.semmelweis-univ.hu (A.M.); 3National Institute of Hematology and Infectious Diseases, Central Hospital of Southern Pest, 1097 Budapest, Hungary; gmikala@dpckorhaz.hu; 4Division of Pathology, Department of Laboratory Medicine, Karolinska Institutet, Karolinska University Hospital, 171 77 Stockholm, Sweden

**Keywords:** follicular lymphoma, circulating tumor DNA, liquid biopsy, *EZH2*, droplet digital PCR

## Abstract

Recent advances in molecular technologies enable sensitive and quantitative assessment of circulating tumor DNA, offering a noninvasive disease monitoring tool for patients with malignant disorders. Here, we demonstrated on four follicular lymphoma cases that circulating tumor DNA based *EZH2* mutation analysis performed by a highly sensitive droplet digital PCR method may be a valuable treatment monitoring approach in *EZH2* mutant follicular lymphoma. *EZH2* variant allele frequencies changed in parallel with the volume of metabolically active tumor sites observed on 18F-fluorodeoxyglucose positron emission tomography combined with computer tomography (PET-CT) scans. Variant allele frequencies of *EZH2* mutations decreased or were eliminated rapidly upon successful treatment, with treatment failure being associated with elevated *EZH2* variant allele frequencies. We also demonstrated spatial heterogeneity in a patient with two different *EZH2* mutations in distinct anatomical sites, with both mutations simultaneously detected in the liquid biopsy specimen. In summary, circulating tumor DNA based *EZH2* mutation analysis offers a rapid, real-time, radiation-free monitoring tool for sensitive detection of *EZH2* mutations deriving from different anatomical sites in follicular lymphoma patients receiving immunochemotherapy.

## 1. Introduction

Follicular lymphoma (FL) is the second most common non-Hodgkin lymphoma (NHL) among adults in developed countries [1,2]. Although the disease has an indolent biology with a median overall survival of 15 years, it is characterized by a remitting, relapsing clinical course with the occurrence of high-grade transformation to a more aggressive lymphoma in around 3% of patients per year [3,4,5,6], generating a need to develop sensitive and broadly accessible tools for disease monitoring. As relapsed and transformed FL is characterized by decreased sensitivity to immunochemotherapeutic regimens, development of new treatment modalities represents an unmet clinical need.

Tumor biopsy represents the cornerstone of the diagnosis of FL, but it is not ideal for disease monitoring due to the invasiveness of the procedure [7]. The process of 18F-fluorodeoxyglucose positron emission tomography combined with computer tomography (PET-CT) is the most widely used method for patient follow-up in lymphoid malignancies [8]; however, it is hampered by its limited sensitivity and specificity [9] and with interpretation of the results being highly dependent on the evaluating radiologist [10]. Liquid biopsy is a novel and emerging radiation-free technique to noninvasively monitor patients with malignant disease and to evaluate their response to treatment [11,12]. Investigating circulating tumor DNA (ctDNA) in patient derived plasma using highly sensitive technologies may overcome the limitations of the other patient evaluating tools mentioned above, as it is less invasive, radiation-free, quick, sensitive and capable of conveying genetic information from a spatially heterogeneous or disseminated tumor [13].

In recent years, several studies focused on ctDNA based treatment monitoring in NHLs, mainly in diffuse large B cell lymphoma (DLBCL). The first two pioneering studies demonstrated that ctDNA based disease monitoring predicts relapse at a median of 3 months earlier compared to imaging based methods [14,15]. Recently, it was found that molecular responses detected with ctDNA analysis after one or two cycles of therapy predict two-year progression free survival (PFS) in DLBCL, defined as early and major molecular responses [16]. However, only limited data is available on the utility of ctDNA based treatment monitoring in FL. The retrospective analysis of pretreatment plasma samples from FL patients enrolled in the PRIMA trial (NCT00140582) demonstrated inferior PFS in patients with higher ctDNA levels [17]. In another study, an elevated amount of pretreatment ctDNA levels correlated with increased baseline total metabolic tumor volume and shorter PFS as determined by PET-CT [18].

Recently, activating mutations of the epigenetic modifier gene, an enhancer of zeste homolog 2 (*EZH2*) in well-defined hotspots of exon 16 and exon 18 (Y646X, A682G and A692V) were described in around 25% percent of FL patients [19,20,21]. *EZH2* represents an attractive therapeutic target in FL, indeed, the United States Food and Drug Administration (FDA) has just granted accelerated approval for the selective EZH2 inhibitor tazemetostat in relapsed/refractory FL [22,23]. In addition to its potential for disease monitoring, noninvasive detection of *EZH2* mutations in the plasma of FL patients could help identify patients who will most likely benefit from *EZH2* targeted therapies in the future.

Here, we present four individual cases of FL, where ctDNA based patient monitoring was performed using a sensitive ddPCR assay for the detection of activating *EZH2* mutations. The obtained mutation profiles were correlated with PET-CT based imaging data where available. Our proof of principle study demonstrates the power of detection and time-resolved monitoring of *EZH2* mutations in ctDNA of FL patients receiving immunochemotherapy.

## 2. Materials and Methods

We analyzed plasma ctDNA samples of four FL patients with known *EZH2* activating mutations determined from formalin fixed paraffin embedded tissue specimens. All patients were treated at the 3rd Department of Internal Medicine, Semmelweis University, Budapest, Hungary. Diagnoses in all cases were based on lymph node histology and immunohistochemistry. For staging and monitoring of the treatment response, PET-CT scans were performed. Routinely, rituximab-based immunochemotherapy was commenced as first line treatment to all patients as described in Appendix A. At first relapse, autologous bone marrow transplant was planned for consolidation, however none of the relapsed patients reached a second remission.

At the time of the first liquid biopsy specimen collection, all patients displayed active disease (FL or transformed FL (tFL)). The number of follow-up liquid biopsies ranged between 1 and 6 per patient. All patients received rituximab in combination with chemotherapy as outlined in Figure 1, Figure 2, Figure 3 and Figure 4. Two patients achieved complete metabolic remission (CMR) with the other two patients succumbing to their disease due to high grade transformation.

Peripheral blood specimens were collected using PAXgene Blood ccfDNA tubes (Qiagen, Germany). Plasma was isolated within six hours in two steps (whole blood was centrifuged at 1.600 g for 20 min at 4 °C, subsequently the plasma fraction was centrifuged at 16.000 g for 10 minutes at 4 °C) following previously published protocols [24,25]. The plasma samples were stored at −70 °C until further processing. ctDNA was isolated using the QIAamp Circulating Nucleic Acid Kit (Qiagen, Germany) following the manufacturer’s instructions. Isolated cell-free DNA samples were stored at −20 °C. Quantity and quality of ctDNA were measured using the Qubit 4 Fluorometer (Thermo Fisher Scientific, USA) and 4200 TapeStation System (Agilent Technologies, USA), respectively.

Mutation specific assays for *EZH2* Y646N, Y646F and *EZH2* A682G (assay IDs: dHsaMDV2516844, dHsaMDV2516772, dHsaMDS555888133, respectively; Bio-Rad Laboratories, USA) were used to evaluate the *EZH2* mutation status with a QX200 ddPCR System (Bio-Rad Laboratories, USA). All reactions were performed using 25 ng of ctDNA. Results were analyzed using the QuantaSoft software (version 1.7; Bio-Rad). All ddPCR reactions were performed with the detection of adequate events (>10,000 droplets per sample) and DNA copies (>10 copies/ul).

All subjects gave their informed consent for inclusion before they participated in the study. The study was conducted in accordance with the Declaration of Helsinki, and the protocol was approved by the Ethics Committee of the Hungarian Medical Research Council (45371-2/2016/EKU).

## 3. Results

### 3.1. Patient #1

A 61 year-old female patient presented with cervical lymphadenopathy in 2012. The histology revealed grade II FL at that time negative for EZH2 mutation (Figure 1). Based on the limited stage IIIA disease, a watch and wait strategy was chosen. In July 2018, the patient developed B symptoms, with the PET-CT scan revealing metabolically active disseminated disease. Inguinal lymph node and duodenal biopsy showed grade II FL and DLBCL, suggesting high grade transformation of the original lymphoma. The ddPCR analysis of these specimens identified an EZH2 Y646N mutation in the inguinal (variant allele frequency, VAF: 26.5%) and an EZH2 Y646F mutation (VAF: 14.6%) in the duodenal biopsy. The analysis of a ctDNA specimen obtained prior to treatment revealed the presence of both Y646N (VAF: 0.4%) and Y646F (VAF: 20.4%) mutations in the bloodstream (Figure 1). The patient received immunochemotherapy (ICT) with R-CHOP (rituximab plus cyclophosphamide, doxorubicin, vincristine and prednisone) as first line treatment. Four weeks later, the second liquid biopsy specimen was found to be an EZH2 wild type, with the subsequent five liquid biopsies collected every 4 weeks also confirming elimination of the EZH2 mutations (Figure 1). The corresponding control PET-CT also demonstrated CMR (Figure 1). To date, the patient is receiving rituximab maintenance therapy with good general health conditions.

### 3.2. Patient #2

A 62 year-old male patient was diagnosed with grade I FL, initially not requiring treatment, in December 2015; the diagnostic tissue sample proved to be EZH2 wild type. In May 2018, disease progression was detected as the PET-CT scan revealed metabolically active areas in the left inguinal and sacral regions (Figure 2). Inguinal lymph node and bone marrow biopsy showed grade II FL and lymphoid infiltration of the bone marrow, respectively, with an EZH2 Y646N mutation detected in both locations with VAFs of 24% and 20.4%. The patient received R-CHOP with rituximab maintenance therapy as first line treatment. In January 2019, the patient relapsed again, and R-DHAP (rituximab plus dexamethasone, high-dose cytarabine and cisplatin) treatment was administered as second line therapy. The liquid biopsy specimen obtained at the time of relapse was positive for the Y646N mutation with a VAF of 31.7%, with the ctDNA sample demonstrating reduction of this mutation (VAF: 1.0%) in response to therapy (Figure 2). In March 2019, R-IGEV (rituximab plus ifosfamide, gemcitabine, and vinorelbine) was started due to clinical refractoriness to the previous treatment line. At this time point, the plasma ctDNA showed an elevated Y646N mutation level (VAF: 39.3%) which was followed by a rapid reduction of the mutant clone (VAF: 1.3%) one week after the initiation of third line treatment. Later, the patient proved to be refractory to all salvage therapies and underwent a palliative surgical intervention due to the growing sacral mass, which caused urinary difficulties. Histological and molecular analyses of the surgically resected area revealed tFL and re-emergence of the previously detected EZH2 Y646N mutation (VAF: 49.4%). The patient deceased in September 2019.

### 3.3. Patient #3

A 62 year-old female patient who presented with generalized lymphadenopathy was diagnosed with an aggressive lymphoproliferative disease, which was proved to be an already transformed follicular lymphoma. Namely, core biopsy of the inguinal lymph node displayed tFL, meanwhile the bone marrow biopsy revealed FL infiltration. In March 2018, the PET-CT scan showed disseminated metabolically active areas. EZH2 A682G mutation was detected in the bone marrow with a VAF of 1.8% (Figure 3). The patient showed no response to first (R-CHOP) and subsequent therapeutic lines (R-DHAP, R-IGEV). Inguinal lymph node biopsy collected in October 2018 reinforced the previous tFL diagnosis and proved to be positive for the original EZH2 A682G mutation (VAF: 38.2%). At this time point, the plasma ctDNA showed an elevated A682G mutation level (VAF: 74.0%). The patient succumbed to the disease in November 2018.

### 3.4. Patient #4

In June 2018, a 73 year-old female patient presented with cubital mass and B-symptoms. A core biopsy of the mass resulted in diagnosis of follicular lymphoma with a 20% proliferation rate. The initial PET-CT described a stage II disease with increased metabolic activity in the right cubital area and in the right axillary lymph node region (Figure 4). At this time, an EZH2 Y646F mutation was detected in the pericubital soft tissue biopsy specimen. Rituximab plus bendamustin (R-B) treatment was commenced as first line therapy. The subsequently analysed ctDNA sample demonstrated reduction of the EZH2 variant down to a VAF of 0.56%. The control PET-CT performed in November 2018 revealed CMR of the disease, with the corresponding ctDNA also found to be the EZH2 wild type. The treatment was completed with local irradiation. Currently, the patient is still in remission with no need of further therapy.

## 4. Discussion

Liquid biopsy-based mutation analysis is becoming a valuable tool for cancer detection and sensitive disease monitoring. In various solid cancers, including lung and colorectal cancer, tumor derived plasma DNA analysis is already incorporated into the routine follow-up and screening guidelines [26,27]. Scrutiny of the clinical applicability of this approach is gaining momentum in hematological malignancies as well, mainly in B-cell disorders. Here, we present four cases of *EZH2* mutant FLs, where ctDNA based treatment monitoring was performed using a highly sensitive ddPCR method to detect *EZH2* mutations in patients treated with ICT. In three out of four cases, the ctDNA based mutation detection was accompanied by parallel PET-CT scan evaluation.

In the field of B-cell lymphomas, the majority of liquid biopsy studies published so far focused on DLBCL. Rossi et al. found liquid biopsy based mutation detection to be eligible for treatment monitoring in DLBCL: in their study, response to therapy was characterized by elimination of the mutations in the peripheral blood, meanwhile treatment refractory disease was associated with the persistence of the previously identified mutations or emergence of novel mutational events [28]. In other studies, two log reduction in ctDNA levels after two cycles of therapy in DLBCL and in Hodgkin lymphoma indicated excellent prognosis [16,29]. In our study, Patient #1 demonstrated a rapid ctDNA clearance upon one cycle of first line treatment which was associated with excellent treatment outcome. Patient #2 had a rapid decrease in *EZH2* Y646N mutant VAF levels after initiation of second- and third-line treatment. The rapid reduction and the subsequent increase of mutant *EZH2* levels can be explained by the cytoreductive effect of chemotherapy and the subsequent emergence of therapy refractory clones. Importantly, this patient has never achieved total clearance of mutant *EZH2* DNA fragments in the plasma which can be linked to inferior outcome. A similar phenomenon was presented in a case report of a patient with colorectal carcinoma where a poor prognosis was observed after incomplete elimination of *KRAS* mutant ctDNA fragments [30]. The occurrence of treatment refractory disease after failure to achieve undetectable *EZH2* mutation levels illuminates the possibility of detecting minimal residual disease (MRD) with the investigation of *EZH2* mutations in the ctDNA of *EZH2* mutant FL patients.

There are only a few studies investigating liquid biopsy based therapy monitoring in parallel with imaging in B-cell lymphomas. Camus et al. found that two thirds of patients with Hodgkin lymphoma presented with negative PET-CT scan results when MRD was detected with liquid biopsy [31]. In FL, Delfau-Larue et al. showed that pretreatment ctDNA levels correlated with the total metabolic tumor volumes measured with PET-CT [18]. In our study, we present three FL cases where ctDNA based treatment monitoring was performed in parallel with PET-CT imaging. In two cases (Patients #1 and #4) elimination of *EZH2* mutations upon treatment was followed by CMR, documented by the PET-CT scans. In these cases, treatment monitoring with liquid biopsy represented a radiation-free, ‘real-time’ therapy monitoring approach compared to PET-CT imaging. In Patient #3, refractoriness to different treatment regimens was associated with persisting metabolically active tumor areas seen on the PET-CT scans and synchronously high VAF of *EZH2* mutated ctDNA was detected. This is in line with previous observations reporting that ctDNA levels correlate with tumor metabolism and aggressiveness in B-cell lymphomas [14,18,32].

As demonstrated by Rossi et al., a single ctDNA specimen is capable of revealing the intra- and intertumoral genetic heterogeneity observed in B cell lymphomas [28]. Recently, we have also reported the applicability of ctDNA based mutation monitoring in unveiling the spatial clonal heterogeneity in the context of therapy resistance in chronic lymphocytic leukemia [33]. In the present study, two different *EZH2* mutations were found in the inguinal FL lymph node and in the duodenal tFL sample of Patient #1; meanwhile, both mutations were detected simultaneously in the liquid biopsy specimen. Indeed, in patients with multiple tumorous areas, a larger portion of the plasma DNA originates from the more aggressive disease site [34,35,36]. Accordingly, in Patient #1, the *EZH2* mutant DNA from the tFL localization was represented by a higher VAF in the liquid biopsy specimen, compared to the *EZH2* mutation originally identified in the lymph node affected by grade II FL (20.4% vs. 0.4%, respectively). Scherer et al. also demonstrated that aggressive lymphomas such as DLBCL and tFL are accompanied by significantly higher ctDNA release compared to FL [32].

Previous studies showed that detection and monitoring of the rearranged immunoglobulin heavy chain gene (*IGH*) is not ideal in FL due to high rates of somatic hypermutation [37]. The utility of the *IgH-Bcl2* translocation in disease monitoring is also controversial, due to the variability of the breakpoint region and presence of this translocation in a proportion of healthy individuals [38,39,40,41]. Previously, several lymphoma-specific hotspot mutations, including *EZH2*, were shown to be detectable from plasma derived ctDNA of FL and DLBCL patients [42,43]. Here, we investigated the feasibility of *EZH2* mutation monitoring in the plasma of FL patients in the context of ICT treatment. In Patients #1 and #4, clearance of *EZH2* mutations was observed upon successful therapy. In Patient #2, *EZH2* mutation allele frequencies changed in parallel with treatment efficacy, while high allele frequency of *EZH2* mutation was detected in the plasma prior to the death of Patient #3. Considering the frequency of *EZH2* mutations in FL, this approach can be applied in approximately 25% of FL cases with a sensitivity of as low as 0.01% depending on the amount of cfDNA. Our findings will most likely gain special importance in the context of *EZH2* targeted therapies currently being evaluated in clinical trials and may serve as a basis for a regular disease monitoring approach in the near future. The selective *EZH2* inhibitor tazemetostat demonstrated objective response rates in 77% of FL patients with *EZH2* mutations in an ongoing phase II clinical trial [44], with *EZH2* mutations detected in both tumor tissue and plasma representing the only predictive biomarker of therapy response [45].

## 5. Conclusions

In our study presenting four cases of EZH2 mutant FL treated with ICT, liquid biopsy based EZH2 mutation analysis offered a rapid, real-time, radiation-free monitoring tool with distinct mutations derived from different anatomical sites detected simultaneously in the ctDNA specimens.

## Figures and Tables

**Figure 1 genes-11-00785-f001:**
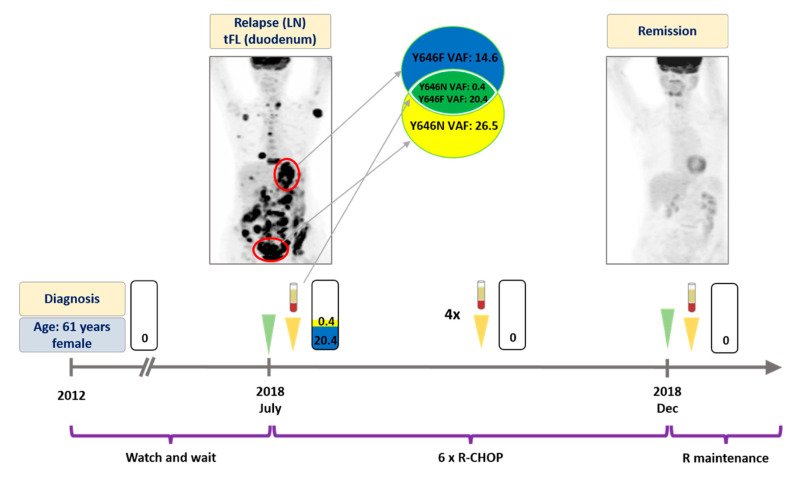
Detailed illustration of treatment monitoring in Patient #1. Spatial heterogeneity was captured in pretreatment plasma and rapid elimination of *EZH2* Y646N and Y646F mutations was observed upon successful R-CHOP treatment. Since then the patient has been receiving rituximab maintenance therapy with good general health conditions. Numbers in the rectangles represent the VAF of the mutations, yellow color and blue colors indicate Y646N and Y646F variants, respectively. Green arrows indicate the time when the PET scans were performed, yellow arrows indicate the time when liquid biopsy specimens were collected. LN: lymph node. tFL: transformed follicular lymphoma. VAF: variant allele frequency. R-CHOP: rituximab, cyclophosphamide, doxorubicin, vincristine, prednisone. R: rituximab.

**Figure 2 genes-11-00785-f002:**
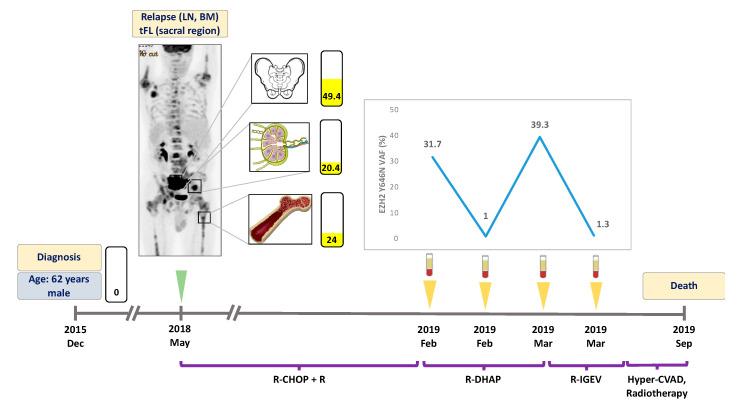
Detailed illustration of treatment monitoring in Patient #2. *EZH2* Y646N mutation was found in three sites at relapse. ctDNA *EZH2* Y646N allele frequencies fluctuated in parallel with treatment efficacy and refractoriness. Failure to eliminate *EZH2* mutations from the plasma indicated poor prognosis. Numbers in the rectangles represent the VAF of the mutation, yellow color indicates the Y646N variant. Green arrow indicates the time when the PET scans were performed, yellow arrows indicate the time when liquid biopsy specimens were collected. LN: lymph node, BM: bone marrow, tFL: transformed follicular lymphoma. VAF: variant allele frequency. R-CHOP: rituximab, cyclophosphamide, doxorubicin, vincristine, prednisone. R-DHAP: rituximab, dexamethasone, high-dose cytarabine, cisplatin. R-IGEV: rituximab, ifosfamide, gemcitabine, vinorelbine. Hyper-CVAD: Course A: cyclophosphamide, vincristine, doxorubicin, dexamethasone. Course B: methotrexate, cytarabine.

**Figure 3 genes-11-00785-f003:**
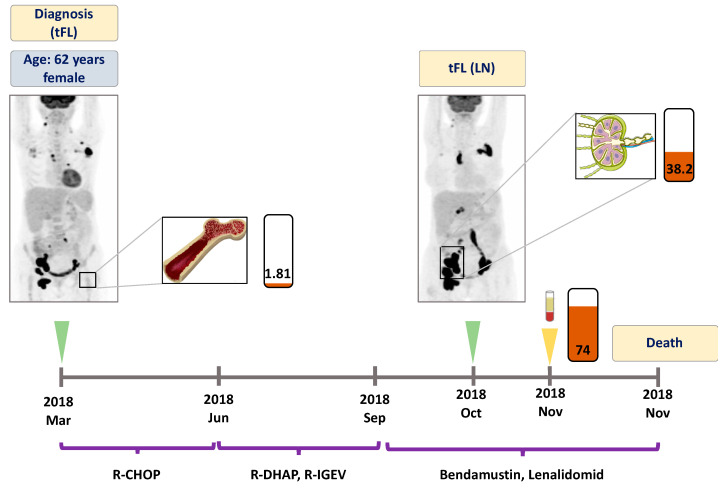
Detailed illustration of treatment monitoring in Patient #3. Liquid biopsy obtained one week prior to the death of the patient revealed high allele frequency of *EZH2* A682G mutation. Numbers in the rectangles represent the VAF of the mutation, brown color indicates the A682G variant. Green arrows indicate the time when the PET scans were performed, the yellow arrow indicates the time when liquid biopsy specimens were collected. LN: lymph node, BM: bone marrow, tFL: transformed follicular lymphoma. R-CHOP: rituximab, cyclophosphamide, doxorubicin, vincristine, prednisone. R-DHAP: rituximab, dexamethasone, high-dose cytarabine, cisplatin. R-IGEV: rituximab, ifosfamide, gemcitabine, vinorelbine.

**Figure 4 genes-11-00785-f004:**
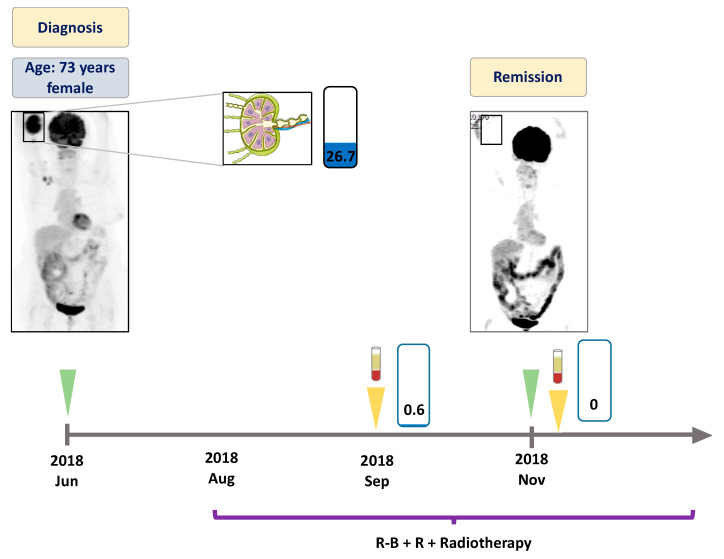
Detailed illustration of treatment monitoring in Patient #4. Low variant allele frequency of *EZH2* Y646F mutation was found in the plasma after one week of treatment initiation. End of treatment ctDNA at three months was found to be *EZH2* wild type, with the corresponding PET-CT revealing complete metabolic response of the disease. Since then the patient has received rituximab maintenance therapy with good general health conditions. Numbers in the rectangles represent the VAF of the mutation, blue color indicates the Y646F variant. Green arrows indicate the time when the PET scans were performed, yellow arrows indicate the time when liquid biopsy specimens were collected. R-B + R: rituximab, bendamustin plus maintenance rituximab.

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
