# Peer review of "Quantitative Analysis and Monitoring of EZH2 Mutations Using Liquid Biopsy in Follicular Lymphoma"

_genes, 2020, doi:10.3390/genes11070785_

Round 1
Reviewer 1 Report
Very interesting manuscript, well written and nicely illustrated
I would suggest to discuss also the limitations of this approach: proportion of follicular lymphoma cases carrying on EZH2 mutations, sensitivity,...
Author Response
Point 1: I would suggest to discuss also the limitations of this approach: proportion of follicular lymphoma cases carrying on EZH2 mutations, sensitivity...
Response 1: Thank you very much for this suggestion. We updated the discussion section with the following statement (lines 270-272): Considering the frequency of EZH2 mutations in FL, this approach can be applied in approximately 25% of FL cases with a sensitivity of as low as 0.01% depending on the amount of the ctDNA.
Reviewer 2 Report
ctDNA characterization is a powerful method to identify tumor-specific genetic aberrations using peripheral blood testing. In this manuscript Nagy et al. have monitored the presence of lymphoma-associated gain-of-function mutations in EZH2 in ctDNA in parallel with imaging of four FL patients. Although the authors found a correlation between variant allele frequencies of mutant EZH2 and response to therapy, the identification of mutations particularly in EZH2 would be more relevant when patients receive EZH2 targeted therapy rather than ICT, as the authors pointed out in the Discussion.
Although not necessary for this manuscript, it would be useful to check for other frequent mutations, especially in the tFL case, to evaluate if new mutations that were not identified at diagnosis are then detected in ctDNA.
Because cfDNA molecules have a short half-life and their concentrations in plasma are low, authors should describe in detail in Materials and Methods how each sample was collected and processed to obtain ctDNA. As supplementary data they should also provide a figure showing the quality of ctDNA.
Author Response
Point 1: Although the authors found a correlation between variant allele frequencies of mutant EZH2 and response to therapy, the identification of mutations particularly in EZH2 would be more relevant when patients receive EZH2 targeted therapy rather than ICT, as the authors pointed out in the Discussion.
Response 1: We acknowledge that treatment monitoring using EZH2 mutation detection using liquid biopsy would be more relevant in the context of targeted treatment. EZH2 inhibitor tazemetostat is currently being evaluated in a clinical trial, once approved, we believe liquid biopsy-based treatment monitoring of EZH2 mutations will contribute to accurate patient follow up. In this paper, our intention was to demonstrate the applicability of the approach in monitoring of FL patients treated with ICT in expectation of the approval of EZH2 targeted therapies and access to samples from patients treated with this modality.
Point 2: Although not necessary for this manuscript, it would be useful to check for other frequent mutations, especially in the tFL case, to evaluate if new mutations that were not identified at diagnosis are then detected in ctDNA.
Response 2: We agree with this highlighting that evaluation of other mutations especially in relapse or transformation setting would be useful in these patients. We are in the process of designing a large, NGS based panel to comprehensively analyze relapsed and transformed FL cases in an independent study.
Point 3: Because cfDNA molecules have a short half-life and their concentrations in plasma are low, authors should describe in detail in Materials and Methods how each sample was collected and processed to obtain ctDNA. As supplementary data they should also provide a figure showing the quality of ctDNA.
We are grateful for this issue raised by the reviewer. We updated the Materials and Methods section with a more detailed preanalytical description of the ctDNA isolation with the following paragraph (lines 95-98): “Peripheral blood specimens were collected using PAXgene Blood ccfDNA tubes (Qiagen, Germany) and within six hours plasma was isolated in two steps (whole blood was centrifuged at 1.600g for 20 minutes at 40C, subsequently plasma fraction was centrifuged at 16.000g for 10 minutes at 40C) following previously published protocols [24,25].”
The quality of all liquid biopsy specimens measured was assessed with 4200 TapeStation System (Agilent Technologies, USA). Purity of the samples was determined by the fraction of small DNA fragments (<500 base pairs) in the specimen. The average ctDNA concentration was 3.3 ng/ul (range: 0.1 – 25.4 ng/ul) as determined by the Qubit 4 Fluorometer (Thermo Fisher Scientific, USA)). Based on the number of detected droplets, the average sensitivity of EZH2 measurements was 0.18% (range: 0.02% - 0.69%).
We also attach a figure for the reviewer presenting the TapeStation quality profiles of each sample analyzed in the study.

Reviewer 3 Report
Nagy et al report four FL cases followed with EZH2 mutations using liquid biopsies. In three cases, post-treatment PET was also available. The data are well presented do not add very much to data available in other lymphoma types since patients are treated with immunochemotherapy and not chemo-free regimens. Also, there in no case the liquid biopsy helps to solve uncertain results at PET.
Author Response
Response 1: In the field of B-cell lymphomas treatment monitoring with liquid biopsy is an extensively researched area with regards to diffuse large B-cell lymphoma. However, studies published thus far in FL focused on the prognostic relevance of liquid biopsy, with no treatment monitoring data available in the literature. In our study, we investigated the power of liquid biopsy-based treatment monitoring in FL patients treated with immunochemotherapy in expectation of the approval of EZH2 targeted therapies and access to samples from patients treated with this modality. There were no uncertain PET results in this cohort consisting of four FL patients, but in one case (Patient #1) elimination of the mutation in the liquid biopsy specimens preceded the documented complete metabolic remission observed in the PET scans by 4 months, which implies that the liquid biopsy approach may reflect the dynamic changes upon treatment.
Reviewer 4 Report
Thank you for giving me the opportunity to review this manuscript.
The authors tried to establish monitoring of EZH2 mutations in ctDNA of FL patients receiving treatment as tool to predict response to treatment. Therefor they analyzed plasma ctDNA samples of four FL patients with known EZH2 mutation. The changes of EZH2 mutation level was compared with response to treatment in PET.
Author tested the 1st pt by diagnosis for EZH2 mutation and he was WT initially, then by PD and transformation in DLBCL pt was positive for the mutation. In the 2nd case it was not mentioned in the text, whether case was also tested at initial diagnosis (pt was diagnosed 2012 and required treatment in 2018).
Green and yellow arrow in the cartoons are not explained.
You may illustrate changes in level of mutation in diagram with adding the treatment times on the curves in all 4 pts.
In table attached, grade is not written by case 4.
Best regards
Author Response
Point 1: Author tested the 1st pt by diagnosis for EZH2 mutation and he was WT initially, then by PD and transformation in DLBCL pt was positive for the mutation. In the 2nd case it was not mentioned in the text, whether case was also tested at initial diagnosis (pt was diagnosed 2012 and required treatment in 2018).
Response 1: We are grateful for this suggestion. Patient #2 was diagnosed in 2015, his diagnostic tumor tissue sample proved to be EZH2 wild type with droplet digital PCR. We updated the result section in line 139: “A 62 years old male patient was diagnosed with grade I FL initially not requiring treatment in December 2015, the diagnostic tissue sample proved to be EZH2 wild type.” In Figure 2, the empty rectangle next to the “Diagnosis” box indicates the EZH2 wild type specimen (with 0% variant allele frequency).
Point 2: Green and yellow arrow in the cartoons are not explained.
Response 2: We thank the reviewer for suggesting this clarification. Yellow arrows indicate the time when the liquid biopsy specimens were collected, green arrows indicate the time when the PET scans were performed. Figure legends have been updated accordingly. Minor correction in Figure 1 was also applied (green arrow in the diagnosis timepoint was removed hence PET scan result is not available at that time).
Point 3: You may illustrate changes in level of mutation in diagram with adding the treatment times on the curves in all 4 pts.
Response 3: In the figures, numbers in the rectangles represent the level (VAF) of the mutations, all mutation types are represented by a unique color. Purple braces at the bottom of the figures represent the specific treatment lines administered to the patient between the indicated timepoints.
Point 4: In the table attached, grade is not written by case 4.
Response 4: Patient #4 was presented with a subcutaneous cubital mass which revealed to be an extranodal diffuse follicular lymphoma variant. According to the current WHO guidelines, this rare type of FL can not be graded. We updated the supplementary table S1 with this additional information.